# A porcine brain-wide RNA editing landscape

Jinrong Huang[1,2,3 ✉], Lin Lin [3,4], Zhanying Dong[1], Ling Yang[1], Tianyu Zheng[1], Weiwang Gu[5], Yan Zhang[6], Tailang Yin[6], Evelina Sjöstedt[7,8], Jan Mulder[7], Mathias Uhlén [7,8], Karsten Kristiansen [2], Lars Bolund[1,3] & Yonglun Luo [1,3,4 ✉]

Adenosine-to-inosine (A-to-I) RNA editing, catalyzed by ADAR enzymes, is an essential post-transcriptional modification. Although hundreds of thousands of RNA editing sites have been reported in mammals, brain-wide analysis of the RNA editing in the mammalian brain remains rare. Here, a genome-wide RNA-editing investigation is performed in 119 samples, representing 30 anatomically defined subregions in the pig brain. We identify a total of 682,037 A-to-I RNA editing sites of which 97% are not identified before. Within the pig brain, cerebellum and olfactory bulb are regions with most edited transcripts. The editing level of sites residing in protein-coding regions are similar across brain regions, whereas region-distinct editing is observed in repetitive sequences. Highly edited conserved recoding events in pig and human brain are found in neurotransmitter receptors, demonstrating the evolutionary importance of RNA editing in neurotransmission functions. Although potential data biases caused by age, sex or health status are not considered, this study provides a rich resource to better understand the evolutionary importance of post-transcriptional RNA editing.

[1] Lars Bolund Institute of Regenerative Medicine, Qingdao-Europe Advanced Institute for Life Sciences, BGI-Qingdao, BGI-Shenzhen, Shenzhen, China. [2] Laboratory of Genomics and Molecular Biomedicine, Department of Biology, University of Copenhagen, Copenhagen, Denmark. [3] Department of Biomedicine, Aarhus University, Aarhus, Denmark. [4] Steno Diabetes Center Aarhus, Aarhus University Hospital, Aarhus, Denmark. [5] School of Biotechnology and Health Sciences, Wuyi University, Jiangmen, China. [6] Department of Clinical Laboratory, Renmin Hospital of Wuhan University, Wuhan, Hubei, China. [7] Department of Neuroscience, Karolinska Institutet, Stockholm, Sweden. [8] Science for Life Laboratory, Department of Protein Science, KTH-Royal Institute of Technology, Stockholm, Sweden. ✉email: huangjinrong@genomics.cn; alun@biomed.au.dk

The brain is the most complex organ in mammals. Recently, we investigated the brain-wide transcriptomics among human, pig, and mouse brain[1]. Apart from transcription, posttranscriptional RNA modifications (PTMs) contribute to expanding function and diversity of transcripts. One of these PTMs is RNA editing, which increases biologically relevant diversity of transcripts or protein isoforms[2]. The most common type of RNA editing in mammals is adenosine-to-inosine (A–I) editing, which is catalyzed by adenosine deaminase acting on RNA (ADAR) enzymes[3,4]. RNA-editing events modulate brain physiology within mammalian central nervous system, especially functions related to neurotransmission[5]. Dysregulation of RNA-editing process has been reported to be associated with several human neurological disorders, such as amyotrophic lateral sclerosis (ALS), autism spectrum disorder, and schizophrenia[6–8]. Some RNA-editing sites have evolved for indispensable PTM in mammalian development. Early postnatal death is observed in the *Adar2*-null mice unable to edit an mRNA transcribed from the *Gria2* gene encoding an AMPA (α-amino-3-hydroxy-5-methyl-4-isoxazolepropionic acid) glutamate receptor subunit[9].

The domestic pig (*Sus scrofa*) shared a common ancestor with human about 79–97 million years ago[10]. Compared to rodents, pigs are more similar to humans in respect of anatomy, genetics, and physiology[11]. Our previous reported transcriptomics analysis revealed that protein-coding gene expression profiles in pig and human brain are highly conserved[1]. In addition, pig models of human diseases can well recapitulate the pathophysiology and symptoms in humans[12].

A recent study analyzed RNA editing using samples from the Genotype-Tissue Expression (GTEx) project, representing 13 subregions in the human brain[13]. However, brain-wide analysis of RNA editing within the mammalian brain remains scarce, making it difficult to better elucidate its functions in the central nervous system. Here we present a porcine brain-wide landscape of A–I RNA editing across 30 brain subregions, organized into 12 main regions, using RNA sequencing (RNA-seq) in combination with whole-genome DNA sequencing (WGS). The regional differences and cross-species (pig–human) similarities in RNA editing were investigated across brain regions.

## Results

### Characterization of genome-wide A–I RNA-editing landscape in pig brain

Two male and two female adult pigs (Bama mini pig, 1 year old) were used for brain-wide RNA-editing investigation. We performed ribosomal RNA-depleted RNA-seq of 119 samples from 30 anatomically defined subregions within pig brain (Supplementary Data 1). These data were generated as a part of our previous creation of the mammalian brain atlas[1]. An average of 173 million uniquely mapped RNA reads were selected for RNA-editing analysis. Further, WGS was performed on individual pig with an average depth of 104× (Supplementary Data 2). By combination of RNA-seq and WGS, we were able to filter out heterozygous single-nucleotide polymorphisms in the genome and accurately identify RNA-editing sites. To investigate the consistency between different procedures for de novo RNA-editing calling, two tools—REDItools[14] and RES-Scanner[15]—were used for comparisons. A high consistency with the results from the two tools was observed (Supplementary Fig. S1). For further investigations, RES-Scanner was used for RNA-editing identification with a pipeline as shown in Supplementary Fig. S2.

We focused on A–I editing, which is the most abundant type of RNA editing in mammals including pig (Supplementary Fig. S3). Our analysis identified 682,037 A–I RNA-editing sites within the pig genome (Supplementary Fig. S2 and Methods).

Approximately 70% of A–I editing sites were located in protein-coding genes. Notably, the majority of sites were located in introns (67.4%), followed by 2.58% in 3′-untranslated regions (3′-UTRs), 0.28% in 5′-UTRs, and 0.25% in coding sequences (CDS) (Fig. 1a). These finding are in line with previously reported RNA-editing sites identified in mouse[16] and human[17], located in the short interspersed nuclear elements (SINEs) within UTRs, introns, and intragenic regions. In agreement with these findings, up to 94% of edited sites identified in the pig brain were within repetitive elements. Further characterization of editing sites in repetitive elements revealed that 82.79% of all A–I editing sites were located in glutamic acid transfer RNA-derived SINEs (SINE/tRNA) (also known as porcine repetitive element, PRE), followed by 8.47% in the LINE1 elements and 2.74% in other repetitive elements (Fig. 1b), similar to previous observations in peripheral pig tissues[18]. The SINE/tRNA elements harboring by far the largest number of editing sites was Pre0_SS, followed by PRE1f, PRE1f2, PRE1g, and PRE1e (Fig. 1c). In line with previous observations, most of RNA-editing sites were at low editing levels (Fig. 1d).

We compared A–I RNA-editing sites identified in this study with those of the PRESDB database, a database of porcine RNA-editing sites from 11 pig organs[19]. Only 18,556 (~3%) editing sites appeared in the PRESDB brain dataset. Hence, >97% of the editing sites identified in this study are previously unknown, including 1687 sites located in CDS and 11,281 sites in 3′-UTRs, respectively. The editing sites identified in this study largely extended the current knowledge of editing landscape in pigs.

Previous studies in other species, such as ant[20] and mouse[16], reported that the density of RNA editing in genes is not random. To investigate the clustering tendency, editing sites were considered as sites in a cluster if more than three sites occurred in a 100 bp sliding window, as described in Li et al.[20]. Although the editing sites located in 5′-UTRs and CDS were similar in number, CDS-residing sites showed a low clustering tendency (Fig. 1e). A high clustering tendency was observed in 3′-UTRs (Fig. 1e). Although much more editing sites in SINE/tRNA were found than in LINE1 elements, the clustering tendency of these two repetitive elements was similar (Fig. 1f).

In addition, it has been found in other species that there is a certain preference of sequence context flanking the editing sites[16,21,22]. To investigate if such a preference of sequence context is also evolutionarily conserved in pigs, we calculated the frequency of bases (A, T, C, G) in the 5 bp regions flanking the editing sites. As compared to randomly selected "A" sites from the genome, there is a significant depletion of G in the −1 base position and enrichment of G in the +1 base position (Fig. 1g), consistent with the known sequence signature of mammalian ADAR enzymes[23]. These sequence preferences were considered as a potential *cis*-regulatory mechanism of A–I editing[24].

Although editing events in coding exons are rare, these editing events can be of functional importance[25]. In the pig brain, a total of 1734 CDS-residing editing sites were identified, representing 1105 protein-coding genes. Out of 1734 sites, 1107 were nonsynonymous (also known as recoding) (Supplementary Fig. S4a). The top three substitution types were glutamine to arginine (Q-to-R), lysine to arginine (K-to-R), and arginine to glycine (R-to-G) (Supplementary Fig. S4b). Of particular note, the gene with the majority of CDS-residing editing sites (9 synonymous and 14 recoding sites) was *SON*, which encodes a protein that binds to RNA and promotes pre-mRNA splicing. Out of 14 recoding sites, 5 were threonine to alanine (T-to-A) and 3 were glutamic acid to glycine (E-to-G). Many recoding editing events are associated with altered protein function[26]. Gene set enrichment analysis further revealed that the genes with at least one recoding site were functionally enriched in neurotransmission-

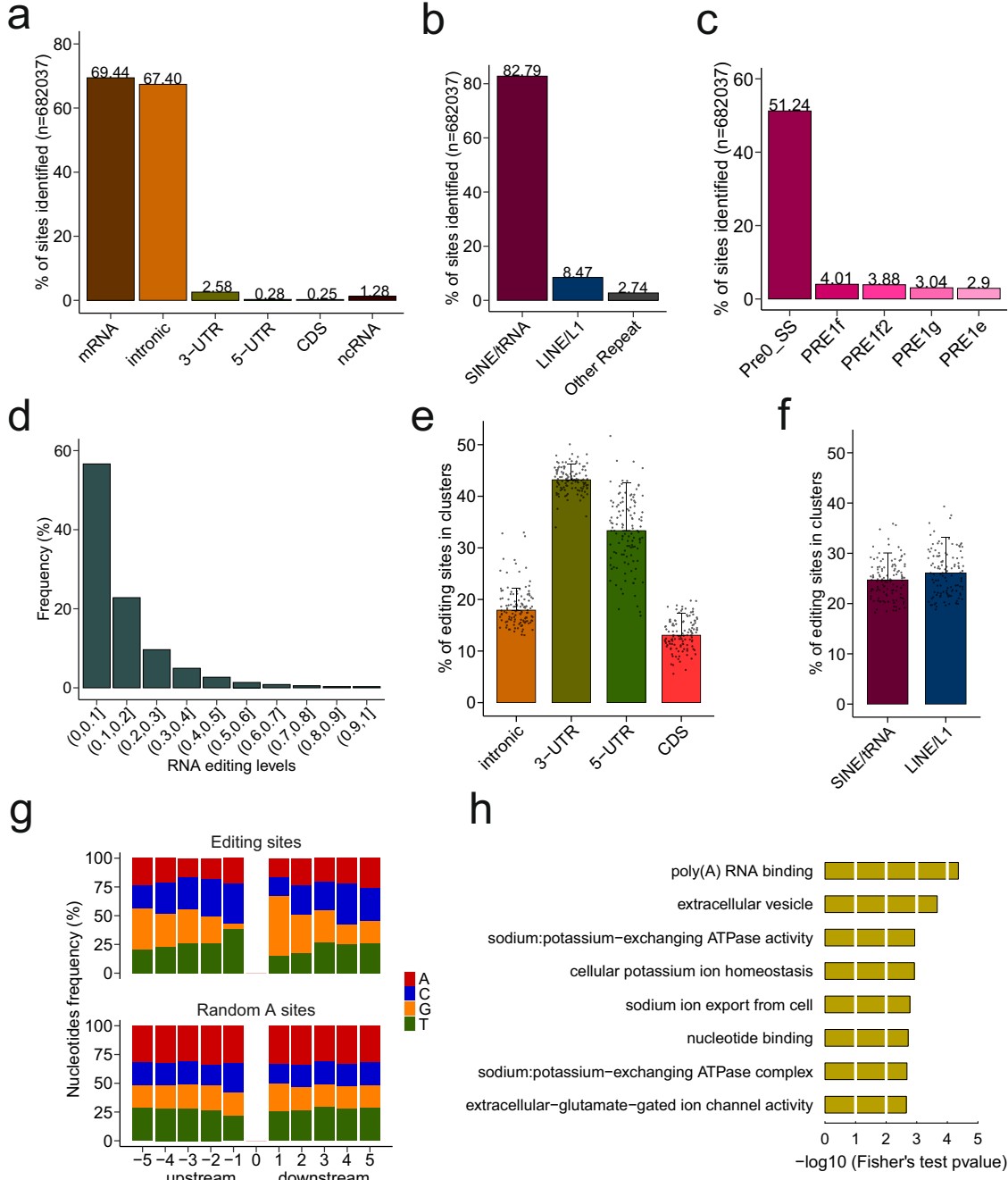

**Fig. 1 A–I RNA editing in the pig brain. a** Proportions of A–I RNA-editing sites across genic regions. **b** Proportions of A–I RNA-editing sites within repeat regions. **c** Proportions of A–I RNA-editing sites within SINE/tRNA elements (only top 5 elements were shown). **d** Distribution of RNA-editing levels. **e** Percentages of editing sites occurring in clusters (at least three sites within a 100 bp window) across intron, 5′-UTR, 3′-UTR, and CDS. **f** Percentages of editing sites occurring in clusters (at least three sites within a 100 bp window) in SINE/tRNA and LINE1 elements. The number of biological replicates is 119. Bar denotes Median + IQR. **g** Frequency of nucleotides in the flanking sequences (5 bp upstream and downstream) of the editing sites and randomly selected genomic "A" sites. **h** The top Gene Ontology (GO) terms associated with genes harboring recoding sites.

related processes, such as glutamate-gated ion channel activity (Fig. 1h), in line with the known role of A–I editing in the mammalian nervous system[27].

Next, 17,581 editing sites were identified in 3′-UTRs. These are potential regulatory sites affecting mRNA stability—e.g., via miRNA interference. Our analysis predicted that 2958 editing sites (17%) in 3′-UTRs altered candidate miRNA-binding sites and 2138 editing sites (12%) might create novel miRNA-binding sites (Supplementary Fig. S4c). Those potential altered editing sites were enriched in genes involved in the mitochondria and

transport-related functions (Supplementary Fig. S4d). Taken together, this provided the first genome-wide characterization of A–I RNA -editing events in pig brain.

**Overall RNA editing across brain regions**. To analyze the brain regional variations in RNA editing, the 30 anatomically defined subregions were organized into 12 regions based on developmental origin or cellular composition (corpus callosum). The number of A–I editing sites identified across 12 brain regions ranged from 240,564 (corpus callosum) to 562,002 (cerebral

cortex) (Supplementary Fig. S5a). We presented the editing sites of each region in *bedGraph* format, which can be visualized by Genome Browser (see PBRe Portal: https://www.synapse.org/PBRe). The number of editing sites shared by all 12 regions was 100,831. It is believed that the number of edited sites identified could increase as more sequencing data are generated[8]. Hence, the number of A–I edited sites were normalized by the uniquely mapped reads of each sample. We found that the normalized number were higher in the cerebellum and olfactory bulb, as compared with other regions (Supplementary Fig. S5a), suggesting that the higher number of edited events in the cerebellum and olfactory bulb is biologically relevant.

To further investigate whether brain regions can be stratified by the RNA-editing landscape, we performed principal component analysis (PCA) based on the sites covered by at least 10 reads in all 30 subregions. The regions from the same brain structure, such as the cerebrum, were clustered together, suggesting that brain regions that are biologically and anatomically similar have closer RNA-editing profile (Fig. 2a and Supplementary Fig. S5b). PCA analysis also showed that the cerebellum is segregated from other brain regions, consistent with a previous study in human[13].

The overall editing levels of CDS-residing edited sites across regions were generally similar, except for corpus callosum (Fig. 2b). However, the overall editing levels of repetitive sites

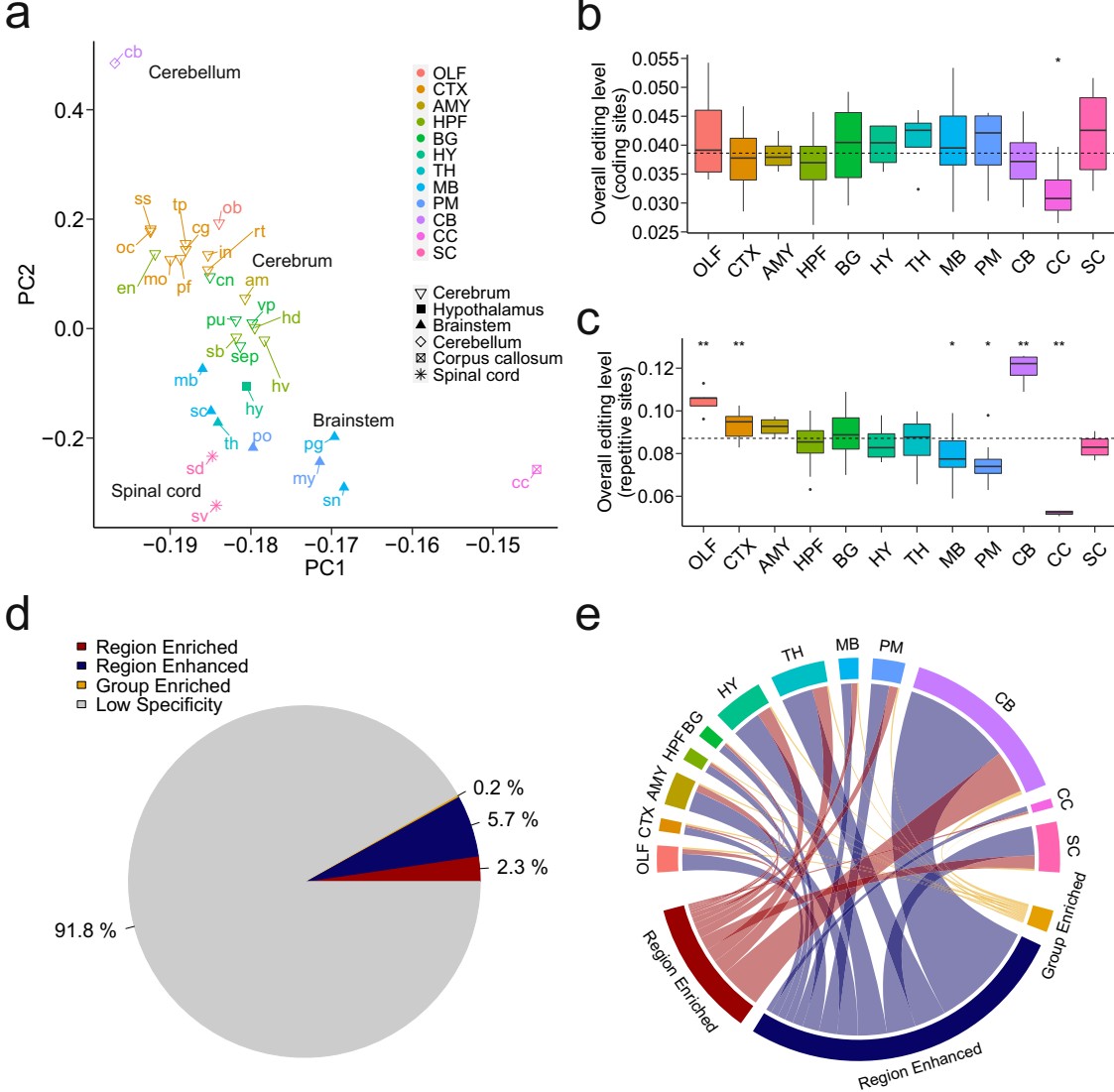

**Fig. 2 RNA-editing profiles across brain regions. a** PCA on editing levels across 30 anatomically defined subregions. The subregions are as follows: olfactory bulb, ob; cingulate cortex, cg; motor cortex, mo; prefrontal cortex, pf; retrosplenial cortex, rt; somatosensory cortex, ss; temporal lobe, tp; insula cortex, in; occipital lobe, oc; amygdala, am; entorhinal cortex, en; hippocampus dorsal, hd; hippocampus ventral, hv; subiculum, sb; caudate nucleus, cn; putamen, pu; septum, sep; ventral pallidum, vp; hypothalamus, hy; thalamus, th; midbrain, mb; periaqueductal gray, pg; superior colliculus, sc; substantia nigra, sn; medulla oblongata, my; pons, po; cerebellum, cb; corpus callosum, cc; spinal cord dorsal, sd; and spinal cord ventral, sv. The 30 anatomically defined subregions are organized into 12 main regions. The regions are as follows: Olfactory bulb, OLF; Cerebral cortex, CTX; Amygdala, AMY; Hippocampal formation, HPF; Basal ganglia, BG; Hypothalamus, HY; Thalamus, TH; Midbrain, MB; Pons and medulla, PM; Cerebellum, CB; Corpus callosum, CC; and Spinal cord, SC. **b, c** Overall editing levels of coding (**b**) or repetitive (**c**) sites in brain regions. The number of biological replicates in each main region is as follows: OLF ($n = 5$), CTX ($n = 31$), AMY ($n = 4$), HPF ($n = 16$), BG ($n = 16$), HY ($n = 4$), TH ($n = 4$), MB ($n = 15$), PM ($n = 8$), CB ($n = 4$), CC ($n = 4$), and SC ($n = 8$). The dashed line denotes the median value of all regions. The significance of differences between overall editing level of each region and that of all regions was assessed by Wilcoxon's test (*$P \leq 0.05$ and **$P \leq 0.01$). **d** The percentage of editing sites classified according to regional specificity. **e** Chord diagrams showing region-specific editing sites across 12 brain regions.

were significantly higher in the cerebellum and olfactory bulb (Wilcoxon's test, $P = 0.0011$ and $P = 0.003$, respectively) (Fig. 2c). The overall editing levels of both CDS-residing and repetitive sites in the corpus callosum were significantly lower than that in other regions, suggesting that the genes expressed in the corpus callosum were less edited (Wilcoxon's test, $P = 0.033$ and $P = 0.0011$, respectively).

To identify edited events that are enriched or shared by one or several regions, the Human Protein Atlas stratification strategy is adapted[1]. We focused on the RNA-editing sites located in genes with low variability of expression (coefficient of variation < 1) and normalized expression ≥ 10 in all main regions. In addition, these sites were required to be covered by at least ten reads in two-third of the main regions. Out of 271,651 editing sites analyzed, 22,275 (about 8.2%) region-specific edited events were identified, including 6359 (2.3%) region-enriched sites, 455 (0.2%) group-enriched sites, and 15,461 (5.7%) region-enhanced sites, respectively (Fig. 2d and see PBRe Portal). The majority of edited events were classified as having low region specificity. The cerebellum harbored the largest number of region-specific edited sites (Fig. 2e). Most of the region-specific editing sites were edited at low levels (Supplementary Fig. S6a). To further investigate the biological importance of region-specific edited sites, Gene Ontology (GO) analysis was performed with genes harboring at least one region-enriched editing site. The genes with low variability of expression and normalized expression ≥ 10 in all main regions were used as background. We identified a series of biological processes and molecular functions specific or shared across brain regions (Supplementary Fig. S6b). For example, the genes with region-enriched sites in pons and medulla were associated with neuron projection development. The genes harboring region-enriched sites in the thalamus were associated with neural-progenitor-specific Brahma related gene 1/Brahma homologue-Associated Factor (BAF) complex. Our results indicate that the region-specific RNA-editing events might play an important role in region-specific neural functions.

Highly edited events, especially those located in CDS or 3′-UTRs, are potentially functional. Thus, we explored CDS and 3′-UTRs residing sites that are highly edited (>75%) in each brain region. GO analysis revealed a number of biological processes and molecular functions specific or shared across brain regions (Supplementary Fig. S7). For example, genes with highly edited sites in the olfactory bulb, cerebral cortex, amygdala, thalamus, midbrain, pons and medulla, and cerebellum were enriched in AMPA glutamate receptor activity.

Furthermore, we examined the expression levels of genes harboring highly (>75%) or lowly edited (<25%) sites. We found that the expression of genes with highly edited CDS-residing sites tended to be lower than that of genes with lowly edited CDS-residing sites (Supplementary Fig. S8a). However, there were no differences in expression between genes with highly and lowly 3′-UTR-residing sites (Supplementary Fig. S8b).

To investigate the relationship between RNA editing and mRNA expression, Pearson's correlation coefficients for genes with normalized expression ≥ 10 in all main regions were calculated. The editing sites ($n = 148,203$) analyzed were required to be covered by at least 10 reads in all main regions. The median correlation coefficients were close to zero, suggesting that there was weak linear relationship for most of RNA-editing sites and genes analyzed (Supplementary Fig. S9).

**Known enzymes responsible for A–I editing**. ADAR1 and ADAR2 are enzymes known to mediate A–I editing in mammals. Both ADAR1 (also known as ADAR) and ADAR2 (also known as ADARB1) were broadly expressed across brain regions, with

lowest expression in the corpus callosum (Fig. 3a). Furthermore, the correlation analysis between expression of mRNA encoding ADARs and editing level of each site was performed. Only the RNA-editing sites (≥10 reads coverage in all main regions) located in genes with low variability of expression (coefficient of variation < 1) and normalized expression ≥ 10 in all main regions were included for analysis. Comparisons of correlation values revealed that a higher correlation was observed between ADAR1 and repetitive sites ($P < 2.22 \times 10^{-16}$), as well as between ADAR2 and repetitive sites ($P < 2.22 \times 10^{-16}$), relative to that between ADARs and coding sites (Fig. 3b). The expression of mRNA encoding ADAR1 explained 22% of the variation in overall editing of all editing sites ($P = 7 \times 10^{-8}$), whereas ADAR2 explained 27% of the variation ($P = 1 \times 10^{-9}$) (Supplementary Fig. S10a, b).

It was reported that the expression of ADAR1 was significantly higher in mature neurons than in glial cells[28]. We extended our analysis with another mRNA expression data set[29], which also supported that the expression of mRNA encoding ADAR1 and ADAR2 is higher in neurons than in non-neuronal cells (Supplementary Fig. S11a, b). It is believed that the cerebellum is a more neuron-rich region in mammals[30]. The expression of mRNA encoding ADAR2 in the pig cerebellum was found to be significantly higher than in other regions (Fig. 3a). Based on these observations, the higher overall editing levels in the cerebellum might be partially explained by the presence of more mature neurons in which ADAR1 and ADAR2 are highly expressed.

ADAR3 is supposed to inhibit RNA editing by competing with ADAR2 for binding to target transcripts[31]. However, the effect of ADAR3 on global editing profiles remains unclear. A number of studies have reported that the expression of ADAR3 has a negative or no association with overall editing levels in human[6,13]. Our analysis revealed that there is no negative correlation between the overall editing levels and the expression of mRNA encoding ADAR3 (Supplementary Fig. S10c).

Recently, aminoacyl tRNA synthetase complex interacting multifunctional protein 2 (AIMP2) is reported to be a negative regulator of active ADARs (ADAR1 and ADAR2)[13]. Our data showed that the expression of mRNA encoding AIMP2 accounted for 4.2% of the variation in overall editing ($P = 0.025$) (Supplementary Fig. S10d). In addition, PIN1 and WWP2 (regulator of ADAR2) were also investigated. However, no association between the overall editing levels and expression of mRNAs encoding these two regulators was observed in the pig brain (Supplementary Fig. S10e, f).

**RNA editing involved in neurotransmission and ion channels**. A–I RNA editing plays important regulatory roles in neurotransmission by altering function or cellular location of neurotransmitter receptors[27]. Here we conducted a survey of RNA-editing events in several major classes of neurotransmitter receptors, including adrenergic, cholinergic, dopamine, GABA, glutamate, glycine, histamine, opioid, and serotonin receptors. The editing sites located in neurotransmitter receptors were mainly found in glutamate receptors (Supplementary Fig. S12a and S13). A small number of recoding sites (Supplementary Fig. S12b) and 3′-UTR-residing sites were detected (Supplementary Fig. S12c). Fifteen recoding sites were located in glutamate receptors of AMPA type (GRIA2, GRIA3, and GRIA4), kainite type (GRIK1 and GRIK2), NMDA (N-methyl-D-aspartate) type (GRIN3B), and metabotropic type (GRM4) (Supplementary Fig. S12b). Some of these recoding sites, such as GRIA2 (Q607R), GRIA2 (R764G), GRIA3 (R775G), GRIA4 (R765G), GRIK1 (Q621R), and GRM4 (Q124R), were considered as evolutionarily conserved in mammals[23].

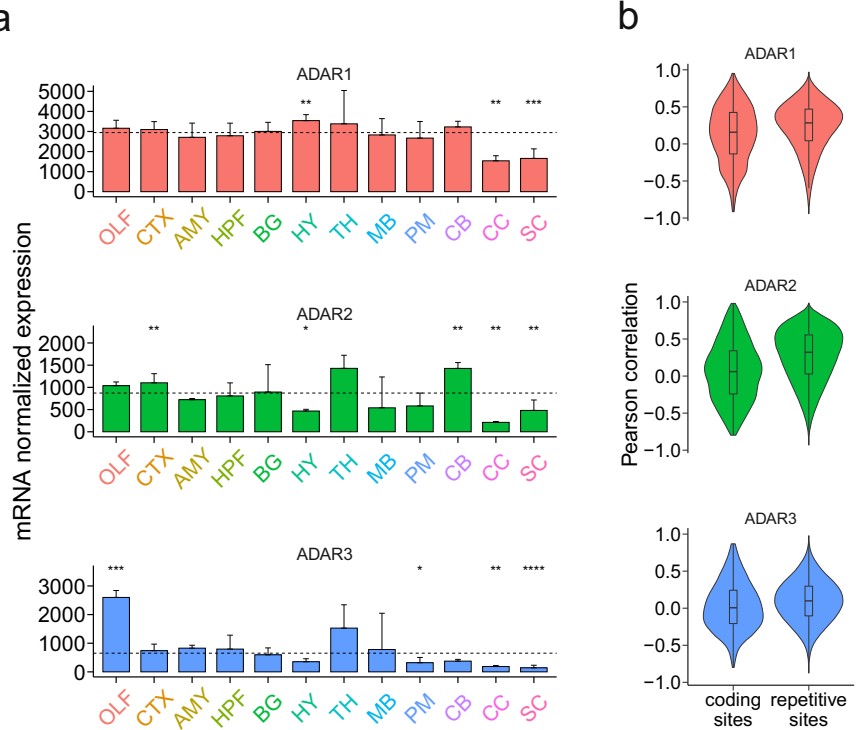

**Fig. 3 ADARs enzymes within the pig brain. a** The mRNA level of ADARs enzymes across brain regions. The number of biological replicates in each main region is as follows: OLF ($n = 5$), CTX ($n = 31$), AMY ($n = 4$), HPF ($n = 16$), BG ($n = 16$), HY ($n = 4$), TH ($n = 4$), MB ($n = 15$), PM ($n = 8$), CB ($n = 4$), CC ($n = 4$), and SC ($n = 8$). Bar denotes Median + IQR for biological replicates in each main region. The dashed line denotes the median value of all regions. The significance of differences between mRNA level of each region and that of all regions was assessed by Wilcoxon test (*$P \leq 0.05$, **$P \leq 0.01$, ***$P \leq 0.001$, and ****$P \leq 0.0001$). **b** Pearson's correlation between mRNA levels of ADARs enzymes and editing levels of non-repetitive coding sites or repetitive sites across brain regions.

The regional similarity and variation in editing level can be observed in these recoding sites (Fig. 4a). Several recoding sites displayed region-specific pattern. For example, the GRIA2 (R764G)-recoding site was region-enhanced in the cerebellum, with 21% increase over the mean editing level of all regions. The increased editing level of GRIA2 (R764G) contributes to the generation of AMPA receptors with faster recovery rates from desensitization[32]. The GRIK2 (Y522C) recoding site had a 21% increase in olfactory bulb over the mean editing level. The GRIN3B (Q208R) editing event was exclusively detected in the spinal cord (Fig. 4a).

AMPA receptors are members of the ionotropic glutamate receptor family and mediate fast synaptic transmission in mammalian brain. Functional AMPA receptors are assembled from GluA1–4 subunits into tetramers[27]. The GluA2 subunit, which is encoded by the *GRIA2* gene, is essential for functional AMPA receptors. Those receptors that consist of GluA2 subunit with edited GRIA2 (Q607R) show low permeability to $Ca^{2+}$, whereas those lacking edited GluA2 subunit show high $Ca^{2+}$ permeability[33]. In a recent mouse model, it has been found that mice with reduced GluA2 Q/R site RNA editing exhibit signs of impairment of several neurological processes, including loss of dendritic spines, hippocampal CA1-neuron loss, learning and memory impairments, and NMDA receptor-independent seizure vulnerability[34]. In the pig brain, we observed that the *GRIA2* transcript contained two recoding sites (Q607R and R764G) and two synonymous sites (Q608Q and L763L) (Fig. 4b). The editing levels of GRIA2 (Q607R)-recoding site across brain regions were close to 100% (Fig. 4a), corroborating that these editing events are indispensable for normal neurotransmission functions.

RNA editing can also regulate the surface expression of some neurotransmitter receptors. One example is type A receptors of

GABA (γ-aminobutyric acid), which is the major inhibitory neurotransmitter in the mammalian brain. Two recoding sites, GABRA3 (I342M) and GABRA3 (N327D), were observed in the pig brain (Fig. 4c). GABRA3 (I342M) was highly edited across all brain regions. A previous study reported that the edited site I342M contributed to the reduction of cell surface expression of GABRA3-containing receptors[35]. The function of the other recoding site GABRA3 (N327D) (9–22% edited) remains to be characterized in future studies (Fig. 4c). Another functionally interesting example is the G protein-coupled serotonin receptor 2C, which is encoded by the *HTR2C* gene. We found that the receptor *HTR2C* transcript harbored seven recoding editing sites (Fig. 4d). Five recoding sites that span residues 156–160 are close to each other and are essential for G protein coupling[36]. These five recoding sites were widely detected across pig brain regions, except for the cerebellum and corpus callosum. An increase of the edited HTR2C isoform was shown to contribute to more efficient cell surface expression of receptors after serotonin stimulation and hence regulate serotonergic signal transduction[36,37]. The other two sites HTR2C (I33M) and HTR2C (T35A) were only found in pig brain, but not in human brain.

In addition, some recoding sites were found in voltage-gated ion channel subunit genes. One example is potassium voltage-gated channel subfamily A member 1 (also known as $K_V1.1$), which is encoded by the *KCNA1* gene and is essential for neuronal excitability. The channel $K_V1.1$ with I400V edited recovers faster from inactivation[27]. A higher (fourfold) editing level of KCNA1 (I400V) was observed in the entorhinal cortex of chronic epileptic rat than in wild type[38]. In the pig brain, editing level of KCNA1 (I400V) varied from 2% to 40% (Fig. 4e). Another example is calcium voltage-gated channel subunit α1D (also known as

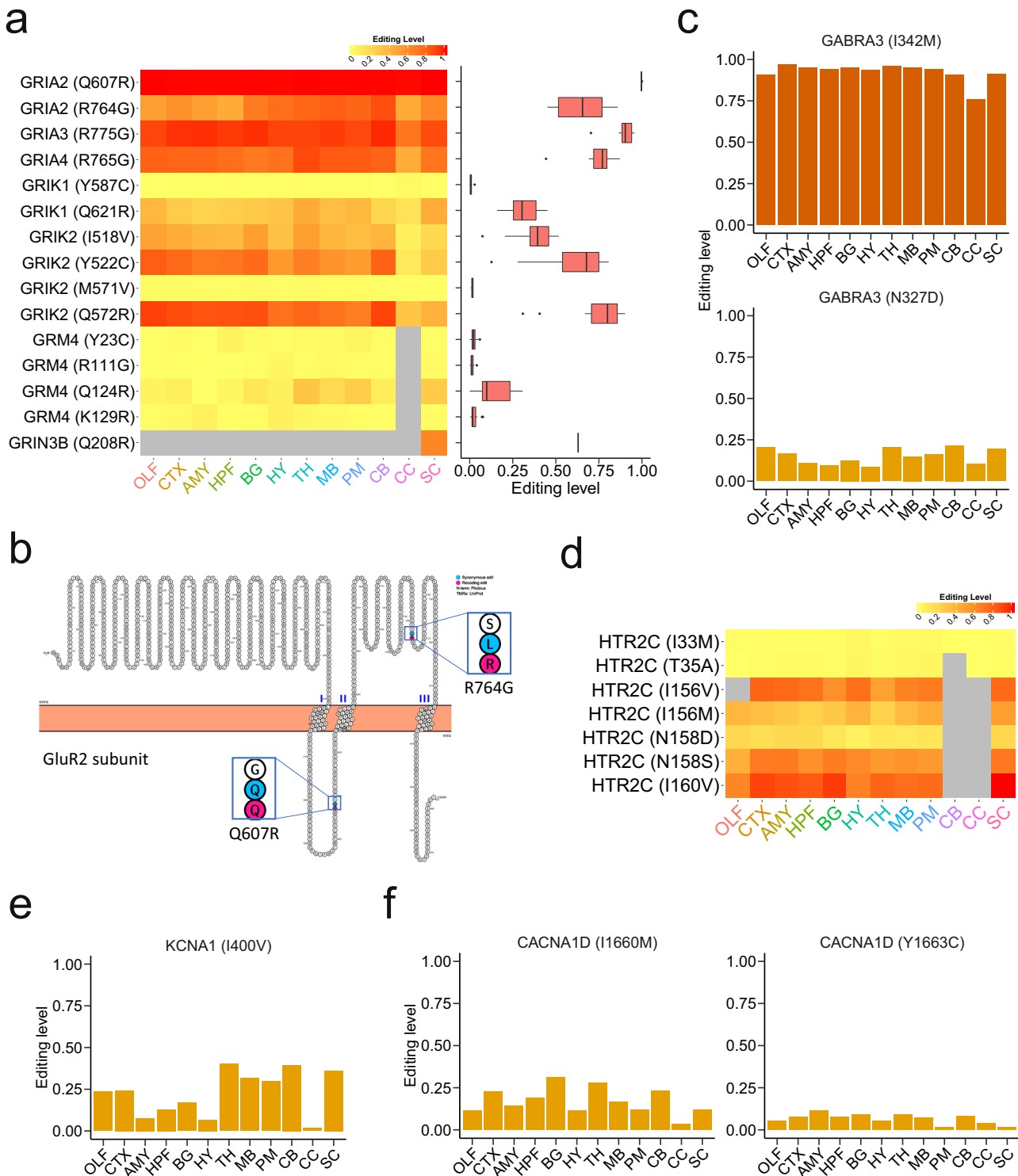

**Fig. 4 A-to-I RNA-editing sites located in neurotransmitter receptors and ion channels. a** Heatmap showing recoding editing sites located in gene encoding glutamate receptors. **b** Four CDS-residing sites located in glutamate ionotropic receptor AMPA type subunit 2. Recoding (nonsynonymous) and synonymous sites are highlighted in deep-pink and deep-skyblue, respectively. **c** Two recoding sites located in γ-aminobutyric acid type A receptor subunit α3 (GABRA3). **d** Heatmap showing recoding editing sites located in G protein-coupled serotonin receptor 2C (HTR2C). **e** One recoding site located in potassium voltage-gated channel subfamily A member 1 (KCNA1). **f** Two recoding sites located in calcium voltage-gated channel subunit α1D (CACNA1D).

$Ca_V1.3$), which is encoded by the *CACNA1D* gene and essential for $Ca^{2+}$ homeostasis. Two recoding sites, CACNA1D (I1660M) (4–31% edited) and CACNA1D (Y1663C) (2–11% edited), were seen in the pig brain (Fig. 4f). Reduced edited $Ca_V1.3$ was observed in the hippocampus of Alzheimer's patient[39].

**Increased editing from fibroblasts to induced neurons**. RNA-editing profiles in pig brain is described above. As the fundamental units of the pig brain, neurons, of which RNA editing remains largely unknown, are there any differences in editing between porcine neurons and non-neuronal cells? To address this

issue, the RNA-seq data from a previous study was included in the analysis[40]. In that study, the porcine fibroblasts were directly converted into induced neurons, which expressed common neuronal markers[40]. Our analysis revealed an increase in overall editing from fibroblasts to induced neurons (Supplementary Fig. S14a), supporting the idea that RNA editing is essential for neuronal function. Two recoding sites Q607R and R764G in GRIA2 were not detected in fibroblasts. As expected, GRIA2 (Q607R) was fully edited (~100%) in induced neurons (Supplementary Fig. S14b). Three recoding sites in GRIK2 (I518V, Y522C, and Q572R) showed an increase in editing in neurons (Supplementary Fig. S14c). *IGFBP7* encodes insulin-like growth factor (IGF) binding protein 7, a member of the soluble proteins that bind IGFs and modulate IGF binding to its receptors[41]. The IGF system is evolutionarily conserved and plays key roles in nervous system function. Elevated editing levels of K95R and R78G sites in IGFBP7 were observed in the neurons (Supplementary Fig. S14d). These two recoding sites map to the insulin growth factor-binding domain of IGFBP7 and potentially generate four transcripts of IGFBP7[42]. The editing of K/E site in cytoplasmic FMR1 interacting protein 2 was elevated in induced neurons (Supplementary Fig. S14e). Elevated editing of S/G site in Neuro-oncological ventral antigen (NOVA) alternative splicing regulator 1 (NOVA1) was also observed (Supplementary Fig. S14f). NOVA1 is a brain-specific splicing factor and RNA editing promotes its stability[5].

**Cross-species analysis between pig and human**. To investigate similarities and differences in RNA editing between pig and human, a total of 770 conserved A–I editing sites were available for analysis (Supplementary Data 3a–f). The conserved sites only accounted for a small fragment of editing sites in pig, consistent with similar findings among other mammals[23]. PCA analysis based on conserved editing sites revealed that brain subregions were grouped by species rather than regions, in accordance with comparisons of human and mouse (Fig. 5a)[13]. This may be partially explained by species-specific RNA-editing regulation. To investigate whether there are any differences in editing levels between conserved and non-conserved sites, we focused on recoding sites. Our analysis revealed that conserved recoding sites were more edited than non-conserved recoding sites in the pig brain (Supplementary Fig. S15), in line with earlier observation in mouse[23].

Furthermore, we investigated genes with conserved and non-conserved recoding sites. Some genes, such as *GRIK1*, *GRIK2*, and *GABRA3*, harbored both conserved and non-conserved recoding sites. Ten genes, including receptor genes (*GRIA2*, *GRIA3*, and *GRIA4*), ion channel-related genes (*KCNMA1*), ion transport-related genes (*UNC80* and *TMEM63B*), and other genes (*ADCY6*, *CPSF6*, *RICTOR*, and *XKR6*), harbored conserved recoding sites solely. The genes with non-conserved recoding sites were mainly in categories related to the nucleus, extracellular exosome, and RNA binding (Supplementary Fig. S16). A number of non-conserved recoding sites were highly edited (>75%) in pig brain (Supplementary Fig. S17). Out of the genes harboring highly edited non-conserved sites, some were potentially important for brain functions. For example, UNC13A and UNC13C are involved in chemical synaptic transmission.

The cross-species comparison of each conserved site was performed across six brain regions, including the cerebral cortex, amygdala, hippocampal formation, hypothalamus, cerebellum, and spinal cord. Most of the conserved sites were unbiasedly edited in the two species (Fig. 5b). Notably, a number of highly edited (>75%) events in both the species were located in genes encoding neurotransmitter receptors, such as *GRIA2*, *GRIA3*,

*GRIK2*, and *GABRA3* (Fig. 5c). This again demonstrated the evolutionary importance of RNA editing in neurotransmission. On the other hand, species-biased edited sites were also observed across brain regions (Supplementary Fig. S18a–f and Supplementary Data 3a–f). For example, GRIK1 (Q621R) was human-biased when edited in the cerebral cortex, hippocampal formation, and hypothalamus. This recoding site has been reported to be less edited in cases of autism spectrum disorder[8]. Two pig-biased recoding sites, IGFBP7 (R78G) and IGFBP7 (K95R), were observed in the cerebral cortex, hippocampal formation, hypothalamus, cerebellum, and spinal cord (Fig. 5d).

In addition, the editing status of recoding sites in genes showing pig-biased or unbiased expression was explored. A total of 122 human RNA-seq read count data obtained from GTEx Portal (Supplementary Data 4) were used for cross-species gene expression analysis. Only the one-to-one orthologous genes ($n = 16,538$) were taken into account. Differential gene expression analysis between pig and human was performed across six brain regions, including the cerebral cortex, amygdala, hippocampal formation, hypothalamus, cerebellum, and spinal cord. The pig-biased genes were defined as genes showing fourfold higher expression in ≥5 regions in pig than in human. No significant difference in recoding editing was observed between pig-biased and unbiased genes (Supplementary Fig. S19).

## Discussion

There are some limitations in this study. First, only a limited number of pig samples ($n = 119$) were included in the study. Future studies with larger sample sizes will increase the statistical power of data analysis. Second, the human subjects aged 20–69 years were included in this study, while the pigs were about 1 year old. Life span varies among species. It is still challenging to determine whether the subjects from different species are at the same age/stage of development. RNA editing and gene expression are believed to be developmentally regulated in mammals[43,44]. When cross-species RNA editing or gene expression comparisons were performed, we did not consider potential data biases caused by differences in age, sex, or health status.

Only 33,779 porcine brain A–I editing sites, which were revealed by RNA-seq data solely, are described in the PRESDB database[19]. Some functionally important editing events, e.g., GRIA2 (R/G) and GRIA4 (R/G), are not described in this database. This may be due to insufficient coverage and sequencing depth of the transcriptome in previous studies. Here we performed a genome-wide RNA-editing investigation across 30 subregions within the pig brain. The combination of RNA-seq and WGS enabled us to de novo identify RNA-editing sites accurately. Consistent with earlier observations in other mammals, A–I editing was the prevalent type of RNA editing in pig. Most of the edited sites identified in this study were previously unknown. Importantly, 1687 novel editing sites located in CDS and 11,281 sites in 3′-UTRs were identified. Besides GRIA2 (R764G) and GRIA4 (R765G), many previously undescribed but potentially important recoding events, such as GRIK1 (Q621R), KCNA1 (I400V), and CACNA1D (I1660M) were described in our PBRe Portal. The CDS-residing sites were small in number and the vast majority of editing sites was located in non-coding regions, such as intron and untranslated regions. We predicted that 29% of editing sites in 3′-UTRs potentially interrupt or create miRNA target sites. Indeed, this study largely expands the existing dataset of porcine brain RNA editing.

In mammals, most A–I editing occur within repetitive elements, especially SINEs, e.g., Alu in human[17], B1 in mouse[16], and PRE in pig (Fig. 1b, c). More and more studies support that A–I editing located in repetitive elements is functionally relevant,

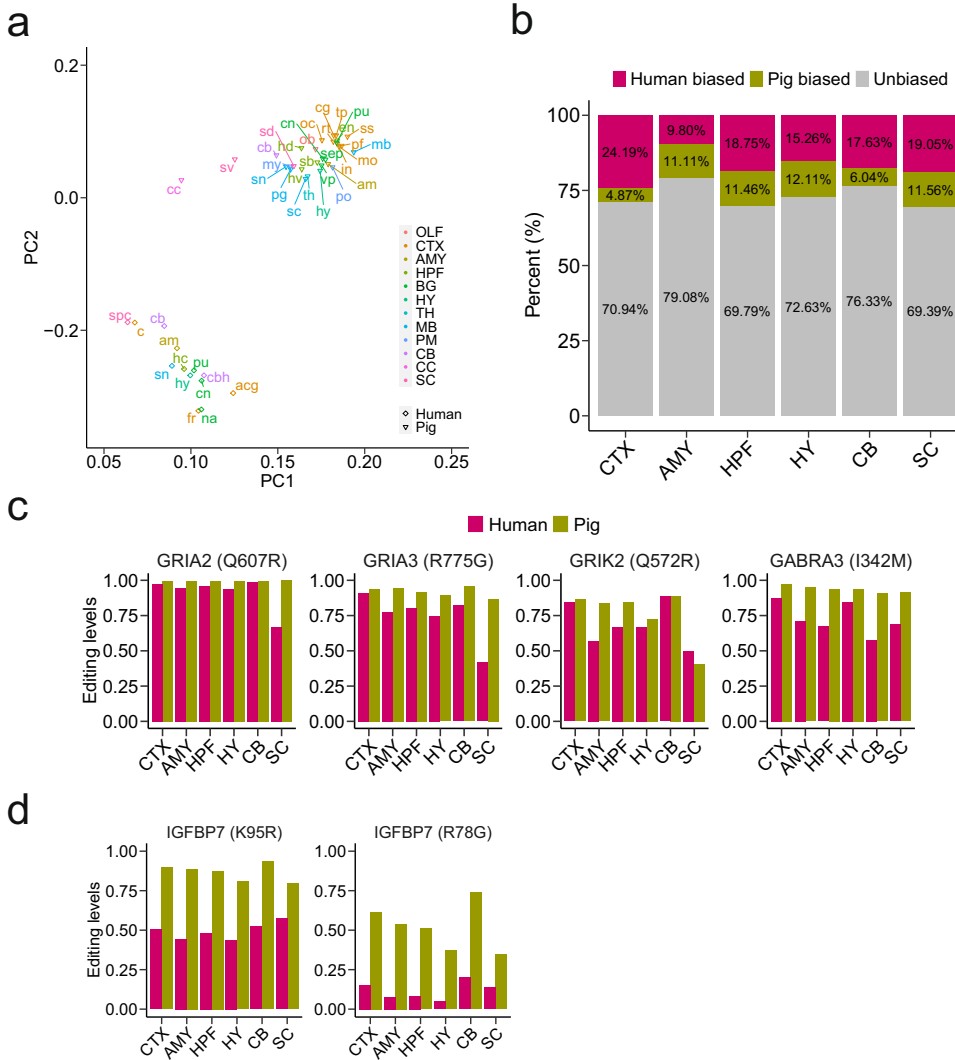

**Fig. 5 Conserved A-to-I editing sites between the pig and human brain. a** PCA on editing levels of conserved editing sites across various brain regions between pig and human. The subregions in human brain are as follows: amygdala, am; anterior cingulate cortex, acg; caudate, cn; cerebellar hemisphere, cbh; cerebellum, cb; cortex, c; frontal cortex, fr; hippocampus, hc; hypothalamus, hy; nucleus accumbens, na; putamen, pu; spinal cord, spc; and substantia nigra, sn. The subregions in pig brain are as follows: olfactory bulb, ob; cingulate cortex, cg; motor cortex, mo; prefrontal cortex, pf; retrosplenial cortex, rt; somatosensory cortex, ss; temporal lobe, tp; insula cortex, in; occipital lobe, oc; amygdala, am; entorhinal cortex, en; hippocampus dorsal, hd; hippocampus ventral, hv; subiculum, sb; caudate nucleus, cn; putamen, pu; septum, sep; ventral pallidum, vp; hypothalamus, hy; thalamus, th; midbrain, mb; periaqueductal gray, pg; superior colliculus, sc; substantia nigra, sn; medulla oblongata, my; pons, po; cerebellum, cb; corpus callosum, cc; spinal cord dorsal, sd; and spinal cord ventral, sv. The subregions are organized into 12 main regions, including Olfactory bulb, OLF; Cerebral cortex, CTX; Amygdala, AMY; Hippocampal formation, HPF; Basal ganglia, BG; Hypothalamus, HY; Thalamus, TH; Midbrain, MB; Pons and medulla, PM; Cerebellum, CB; Corpus callosum, CC; and Spinal cord, SC. **b** Comparisons of conserved editing sites across six brain regions, including the cerebral cortex, amygdala, hippocampal formation, hypothalamus, cerebellum, and spinal cord. **c** A number of highly edited sites in both two species were highlighted. **d** Two pig-biased recoding sites located in insulin-like growth factor-binding protein 7 (IGFBP7).

rather than byproduct of ADARs. SINEs are one source of endogenous double-stranded RNA (dsRNA). The unedited endogenous dsRNA that is similar in structure to viral dsRNA has the potential to trigger the activation of the antiviral innate immune system[2]. One important function of RNA editing is to prevent inappropriate activation of the cellular immune system[45]. In addition, SINEs are one type of active retrotransposons in the mammalian genome. The random insertion of SINEs Alu into the genome has been reported to be associated with genetic disease in humans[46]. Thus, another possible function of editing in repetitive elements may be to alter sequence and affect the integration of retrotransposons back into the genome[4]. However, the biological roles of editing in repetitive elements remain difficult to infer and further investigations are required.

Regional variations in RNA editing have been observed in the human[13] and pig brain (this study), indicating that RNA editing is spatially regulated in the mammalian brain. Similar to observations in human, the overall editing levels of CDS-residing sites tended to be similar across pig brain regions, whereas editing levels of those sites in repetitive region were more likely to be different. This suggested that there were different factors contributing to RNA editing in CDS-residing and repetitive sites. The spatiotemporal expression of ADAR enzymes is a known *trans*-regulatory mechanism of A–I editing. We observed that the overall correlation between expression of ADAR1 and editing level of repetitive sites was higher than that between ADAR1 and coding sites in the pig brain, similar to earlier observations in human[13]. Unexpectedly, there was no high correlation between

the expression of ADAR2 and editing level of coding sites in the pig brain, contrasting the result in human, which is drawn based on thousands of samples from whole body[13]. Considering the small sample sizes used in this study, the differences might be due to lack of enough statistical power. More samples are needed to better address this in future studies.

The expression of ADAR2 was remarkably higher in the pig cerebellum and the editing profile in cerebellum was distinct from those in other brain regions. The overall editing analysis revealed that repetitive sites in the cerebellum were edited more, in line with earlier observations in human[13]. In the mammalian brain, the highest density of neurons (>80%) is seen in the cerebellum[30,47]. The expression of ADAR1 and ADAR2 is considered to be higher in neurons than in other cell types[29]. The distinct editing signature in the cerebellum might be associated with the high proportion of neurons. In-depth investigations, such as more samples and single-cell RNA-seq, will be able to test this hypothesis.

ADAR enzymes are highly conserved in mammals. The number of known A–I editing sites varies among species. For example, according to REDIportal (V2.0)[14], there are 107,094 A–I editing sites (including 224 recoding sites) in mouse, which is much less than that in human or pig. Only a small number of conserved editing sites were found between human and mouse[23], or between pig and human (this study). One similarity among human, mouse, and pig is the fact that many conserved recoding events occur in neuronal genes.

Cross-species editing analysis of pig and human brain revealed that most of the conserved editing sites displayed an unbiased pattern. Of particular note, many physiologically important editing events were unbiasedly edited. For example, out of unbiased edited events found in the cerebral cortex, GRIA2 (Q607R) and GRIK2 (Q572R) are involved in regulating $Ca^{2+}$ permeability[26]. GRIA2 (R764G), GRIA3 (R775G), and GRIA4 (R765G) are involved in receptor desensitization[26]. GABRA3 (I342M) is involved in receptor trafficking[26]. This again indicated the biological importance of RNA editing in the central nervous system. On the other hand, more human-biased sites were observed in some regions, such as the cerebral cortex. An earlier study reported that conserved sites were often edited more in human than in mouse brain[13]. These observations are in line with the fact that the human brain has greater complexity than the brain of other mammals.

It is known that aberrant RNA editing caused by, e.g., altered ADAR activity is associated with many human diseases, such as cancers, metabolic diseases, autoimmune disorders, and neurological disorders[48]. Deficient RNA editing in the AMPA glutamate receptor gene GRIA2 has been found to be associated with the development of ALS in human[7]. To investigate the underlying mechanisms of human disease, mouse is one of the primary model organisms. For example, the functional importance of Q/R site in Gria2 has been demonstrated in the Adar2-null mouse[9]. However, anatomical differences between mouse and human brain are needed to be considered. For example, mouse lacks gyri and sulci in the cerebrum, unlike human or pig. In addition, many genes related to neurotransmission are subject to RNA editing. A higher overall correlation between pig and human brain for expression of genes related to neurotransmission was observed, as compared with that between mouse and human brain[1]. As stated previously, we failed to reveal high correlation between ADAR2 expression and editing of coding sites in the pig brain. However, most of conserved CDS-residing sites, including many physiologically important editing events, were unbiasedly edited between pig and human brain. Based on these observations, pig may be an alternative choice for studying dysregulated RNA editing associated with human neurological disorders.

Taken together, our study provides a valuable resource to understand the complexity of the mammalian brain, as well as broaden the application of pigs in biomedical research.

## Methods

**Sample information for RNA-seq.** The samples were collected from 30 subregions within pig brain (Supplementary Data 1). The detailed animal procedures were described in our previously published paper[1]. Sample collection and handling of animals were carried out in accordance with national guidance for large experimental animals and under permission of the local ethical committee (ethical permission numbers BGI-IRB18135). Four Chinese Bama minipigs (2 male and 2 females, 1 year old), were provided by the Pearl Lab Animal Sci & Tech Co., Ltd, where the animals were housed in a special pathogen-free stable facility under standard condition.

**WGS and mapping.** Genomic DNA was purified from the liver of each pig. The DNA libraries were constructed using MGIEasy Universal DNA Library Prep Kit and sequenced on the BGISEQ-500 with 100 bases paired-end reads. We generated clean data with an average depth of 104× for each pig (Supplementary Data 2). Clean DNA sequencing (DNA-seq) reads were mapped to Sscrofa11 genome (ensembl release-92) with parameter "mem -t 4 -M -Y -R" using BWA (version 0.7.15-r1140)[49].

**RNA-seq mapping and gene counts normalization.** Clean RNA-seq reads were mapped to Sscrofa11 genome (ensembl release-92) with parameter "-q --no-mixed --no-discordant -p 3" by HISAT2 (version 2.1.0)[50]. The gene counts of each sample were calculated using RSEM (version 1.2.12)[51]. For comparisons between samples, the gene counts were normalized by normalization factor (also known as size factor)[52] using the normalize.deseq function in metaseqR package[53].

**Comparison of results from REDItools and RES-Scanner.** To compare the results from REDItools and RES-Scanner, the same set of initially pre-aligned RNA and DNA BAM files were used as input for both tools. Only sites that met the criteria were used for comparisons: (I) homozygous for gDNA and with a coverage of at least ten DNA reads; (II) a coverage of at least ten RNA reads; and (III) at least three edited reads and editing level ≥ 5%.

**Processing of unmapped RNA reads.** Recent studies have reported that numerous RNA reads were missed by regular alignment due to hyper-editing[21,54]. To retrieve these reads, the RNA reads discarded by initial alignment were transformed (As to Gs), re-aligned to transformed reference genome (As to Gs) by HISAT2, and then recovered as described in Porath et al.[54]. The recovered aligned RNA reads in BAM format were used for the subsequent RNA-editing analysis.

**de novo RNA-editing identification.** Two rounds of A–I RNA-editing identification were performed in this study. The first round, also called de novo RNA-editing identification, was performed on each sample to obtain a list of editing sites by RES-Scanner[15]. Together with pre-aligned DNA-seq reads, the initially pre-aligned RNA reads and recovered pre-aligned RNA reads in BAM format of each sample were used as inputs for RES-Scanner. The potential PCR duplications of DNA or RNA reads were discarded using SAMtools (0.1.19)[55]. An average of 173 million uniquely mapped RNA reads were obtained for RNA-editing analysis of each sample. Strict criteria were used as follows: (I) a candidate site was required to be homozygous for gDNA and with a coverage of at least ten DNA reads; (II) any RNA reads with quality score < 30 for the site were discarded. The first and last six bases of each aligned RNA read were clipped. A candidate site should be supported by at least three edited RNA reads with editing level ≥ 5%; and (III) in order to avoid misalignment to paralogous regions, the uniquely mapped RNA reads were re-aligned to the genome by BLAT[56]. The sites with a proportion of qualifying reads to total BLAT-realigned reads < 50% were discarded. (IV) A candidate site was required to be supported by at least two samples. The default parameters of RES-Scanner were used if it is not mentioned specifically. The qualifying sites identified in each sample from different regions were combined to generate a comprehensive list of A–I editing sites in the first round.

Due to the stringent criteria used in the first round, some true-positive editing sites were missed in some samples. For example, the site with low editing level (<5%) or low edited coverage (<3 edited reads supporting) were discarded. Hence, a second round of RNA-editing identification were performed to retrieve such missed editing sites. A more liberal criterion was used: each candidate site was required with editing level > 0 and ≥1 edited RNA read supporting editing. However, at least ten total RNA reads supporting edited and non-edited were required for each site, allowing us to quantify editing levels accurately. Finally, a total of 682,037 A–I RNA-editing sites were identified and an average of 179,927 A–I editing sites were obtained for each sample.

**Editing level calculation.** The editing level of each site was calculated as the number of Gs divided by the total number of As + Gs. The As or Gs of different

samples from the same subregion or region were combined to calculate the editing level of each site at the subregion or region level. The overall editing level of each sample was calculated as the number of Gs divided by the total number of As + Gs at all editing sites.

**miRNA targeting prediction**. A total of 457 known mature miRNA (*S. scrofa*) were downloaded from miRBase database (release 22)[57]. A seed (7 bp) was defined as positions 2–7 of a mature miRNA. To investigate the effect of RNA editing on miRNA targeting, we compared the flanking 13 bp-long sequence of the editing site (6 bp each sides) and the seed of known miRNA. A site is regarded as a candidate binding site if any 7 bp sequence can be completely complementary to a known miRNA seed, as described in Peng et al.[58].

**Previous pig editing resource**. RNA-editing sites of previous study in pig were downloaded from PRESDB database (https://presdb.deepomics.org)[19].

**Region-specific RNA-editing identification**. To investigate region-specific editing, three subcategories, including region enriched, group enriched, and region enhanced, were defined in this study. The editing levels of region enriched, group-enriched, or region-enhanced sites were required to be at least 25%. Region-enriched sites were defined as those with an editing level at least 20% higher in a particular region as compared to any other region. Group-enriched sites were defined as those with an editing level at least 20% higher in more than one but less than one-third of regions (two to four regions in this study) as compared to any other region. Region-enhanced sites were defined as those with an editing level at least 20% higher in a particular region as compared to the average level in all regions.

**The mRNA expression scores of human neurons and non-neuronal cells**. The mRNA expression scores of neurons, microglia, astrocytes, and oligodendrocytes from 20 human brain regions, which are based on co-expression analysis[29], were obtained from http://oldhamlab.ctec.ucsf.edu/. The significance of differences between expression scores of ADAR1 or ADAR2 in neurons and non-neuronal cells was assessed by Wilcoxon's test.

**Processing of porcine fibroblasts and induced neurons RNA-seq data**. The RNA-seq data (including Sequence Read Archive (SRA) data and Transcript per Million (TPM) data) of fibroblasts (*n* = 3) and neurons (*n* = 9) were downloaded from Gene Expression Omnibus using accession number GSE146494[40]. The SRA data format was converted to fastq by fastq-dump in SraToolkit (2.8.2). Then paired-end RNA-seq reads were mapped to Sscrofa11 genome (ensembl release-92) with parameter "-q --no-mixed --no-discordant -p 3" by HISAT2 (2.1.0)[50]. The potential PCR duplications of RNA reads were discarded using SAMtools (0.1.19)[55].

**Known RNA-editing calling**. For porcine fibroblasts and neurons, RNA-editing calling was performed based on sites identified in our study using an in-house script. The uniquely mapped RNA BAM file was used as the input for analysis. Any RNA reads with quality score < 30 for the site were discarded. At least four total RNA reads supporting edited and non-edited status were required for each site.

**Differential RNA-editing analysis between species**. A–I RNA-editing sites in a human brain atlas, including 332 samples, were downloaded from REDIportal (http://srv00.recas.ba.infn.it/atlas/index.html)[59]. To compare editing sites between pig and human, the conserved sites were obtained from the pig and human genome. The coordinates of human editing sites were converted to coordinates on the pig reference genome using the liftOver tool and the "hg19ToSusScr11.over.chain.gz" file from UCSC (http://genome.ucsc.edu). The total number of RNA reads supporting edited and non-edited status in a given conserved editing site were combined from samples corresponding to the same brain region. The differential editing sites at the same brain region between the two species were identified using Fisher's exact test, with difference of editing level ≥20% and *P*-value < 0.01.

**Differential gene expression analysis between species**. The RNA-seq read counts of 122 samples (Supplementary Data 4), covering the human amygdala, cerebellum, cortex, hippocampus, hypothalamus, and spinal cord, were downloaded from GTEx Portal (https://gtexportal.org/home/datasets). Differential expression analysis between species was performed by DESeq2[52]. Only the pig-to-human one-to-one orthologous genes (*n* = 16,538) were included for analysis. Species-specific expressed genes were defined as genes with fold change ≥ 4 and false discovery rate <0.01.

**Principal component analysis**. The editing sites with at least ten RNA reads coverage in all subregions within the pig brain were used for PCA analysis. PCA was performed using the *prcomp* function in R language.

**GO analysis**. We performed GO enrichment analysis using DAVID online Resources[60]. GO terms were defined as significant with Fisher's exact test *P*-value < 0.05.

**Statistics**. Fisher's exact test was used to assess the significance of GO enrichment or differential editing analysis. Wilcoxon's test was used to assess the significance of median differences between two groups. The editing sites identified in samples from the same subregion/region were pooled for further investigations.

**Reporting summary**. Further information on research design is available in the Nature Research Reporting Summary linked to this article.

## Data availability

The RNA-seq and WGS data have been deposited into CNGB Sequence Archive (CNSA)[61] of the China National GeneBank DataBase (CNGBdb)[62] with accession number CNP0000483 and CNP0001045. The analytical results are available through the Pig Brain RNA editing (PBRe) Portal: https://www.synapse.org/PBRe.

## Code availability

The computing scripts developed for genome-wide profiling of RNA-editing events in pigs are shared in GitHub (https://github.com/JinRcn/PigBrainRNAediting). BWA (0.7.15-r1140), HISAT2 (2.1.0), RSEM (1.2.12), SAMtools (0.1.19), RES-Scanner, REDItools (1.3), perl (v5.26.0), Python (2.7.5), R (3.6.3), ggplot2 (3.3.0), gplots (3.0.3), UpSetR (1.4.0), circlize (0.4.8), GenomeGraphs (1.46.0), ggpubr (0.2.5), and ggrepel (0.8.2) were used in this study.

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

## Acknowledgements

We thank China National GeneBank for providing the storage of data. This project is partially supported by the DFF Sapere Aude Starting grant (8048-00072A, to L.L.) and the Sanming Project of Medicine in Shenzhen (SZSM201612074, to L.B.).

## Author contributions

Y.L. and L.B. conceived and designed the project. J.H., Z.D., E.S., J.M., and Y.L. contributed to sample collection. Z.D. and L.Y. contributed to library construction and sequencing. J.H. performed the bioinformatics analysis, prepared the figures, and drafted the manuscript. L.L. and Y.L. supervised the study and data analysis. J.H., L.L., T.Z., W.G., Y.Z., T.Y., E.S., J.M., M.U., K.K., L.B., and Y.L. contributed to data interpretation, text revision, and discussion. All authors contributed to the final manuscript.

## Competing interests

The authors declare no competing interests.
