## [Peer Review File · Communications Biology]

Reviewers' Comments:

Reviewer #1:

Remarks to the Author:

This study revealed the RNA editing events in the pig brain for the first time, and made a good supplement to the research on RNA editing in the mammalian brain. However, existing studies have revealed the situation of RNA editing events in the human brain, which greatly reduces the novelty of the research. The RNA editing sites in some key genes require to confirm their authenticity and functions.

1. There are many softwares that have been published to identify RNA editing events. In addition to RES-Scanner, authors can consider choosing other softwares or processes for RNA editing calling, and then compare the consistency of the results from different processes.
2. What are the effects of the genes with the highest editing level or editing frequency in different regions on regional function, and how is the expression level of these genes?
3. What are the functions of genes with conservative editing sites or with non-conservative sites among species? Is there a difference in editing levels between conservative editing sites and non-conservative editing sites? Does the editing level of conservative editing sites differ among species?
4. How about the RNA editing status of species-specific genes or differentially expressed genes between human and pig?
5. In addition to conserved RNA editing sites, species-specific RNA editing events with high editing level also need attention, because specific high-level editing sites may cause differences in brain function and structure between species. So, please provide more comparisons.
6. It is recommended to verify the authenticity and functions of the RNA editing sites of some important genes.
7. More experiments may be needed to prove that RNA editing plays an important role in pig brains, and more explanations on the role of RNA editing on brain function are also needed.

Reviewer #2:

Remarks to the Author:

This is a highly interesting and very well written resource manuscript covering A-to-I RNA editing sites in 30 anatomically defined subregions of the porcine brain. The RNA editing sites were identified de novo by a combination of RNA-seq and whole genome sequencing. Given the role of altered RNA editing in neurological disorders and the growing importance of the pig as a model organism, this manuscript will receive a lot of attention and help to design translational pig models for neurological diseases. The manuscript is an important complementation to corresponding studies in human and mouse.

Minor: line 351 "Whether" instead of "Weather"

Reviewer #3:

Remarks to the Author:

In their current study entitled "A porcine brain-wide RNA editing landscape" Huang et al., described a porcine brain-wide RNA landscape which provides a rich resource to better understand the evolutionary importance of post-transcriptional RNA editing. They identified a total of 682,037 A-to-I RNA editing sites in 30 different anatomically distinct brain subregions in pig brain. Importantly, 97% of the RNA editing sites were not previously identified and highly conserved recoding events are found in neurotransmitter receptors demonstrating an evolutionary importance of RNA-editing process on brain function. Furthermore, these studies reconfirm A-to-I editing of GluA2 RNA is critical for brain function across different mammalian species. Over all these studies provide a framework to investigate the role of previously unidentified RNA-editing in human brain function. The manuscript is generally well written, with the hypothesis and experimental paradigms well justified and the results of the study not overstated. If the authors are able to provide more clarification, primarily in the discussion section and minor changes in the description of figure legends, statistics used in the methodology section then manuscript will be

suitable for publication.

Specific comments, with recommendations for addressing each comment

- 1) In the text (lines 94-97), the author showed 70 % of editing sites are located within the protein coding sequences. Notably, 67.4% editing sites in the introns. While the finding is very interesting, looking at the Figure 1a, it is hard to understand the % of editing sites in the graph. Is the quantification was determined in relation to total 100% editing sites? Please clarify.
- 2) Identification of clustering tendency of the editing sites is very informative. The author mentioned (lines 116-117) that clustering studies were previously, carried out in ant. It is critical to mention whether this kind of studies were performed in mice or rat.
- 3) Is there in any studies in the literature that suggest clustering if 3 sites are located in a 100bp sliding window (lines 118-119)?
- 4) Please provide the information about statistics in the legend for Figure 2b and 2c. (e.g # biological replicates, # of technical replicates if applicable, statistical test conducted e.g Wilcoxon test, p=values)
- 5) Which expression levels were measured (lines 209-211)? Is it mRNA or protein expression levels? Expression levels were normalized against which reference gene or protein? Kindly, provide the statistical information for graphs of ADAR1, ADAR2 and ADAR3 expression in Figure 3a.
- 6) What are these abbreviations PC1 and PC2 in Figure 5a? Please provide the information in figure legend.
- 7) This figure legends in Figure S1 should describe BWA, HISAT2 instead of using abbreviation only. It will help reader to follow and check the methodology in detail. Also, it would be good to mention the 30 anatomically different regions in the flowchart.
- 8) In Figure S2, please write this is WGS analysis. It would be important to mention which brain regions were analysed in this graph.
- 9) Kindly clarify (line 138, Figure S3b), whether the number of events is an absolute or relative number.
- 10) In the text (lines 220-221/Figure S6) is it mRNA or protein expression levels (Figure S6)? Is this overall A-to-I editing levels? Or of a particular edited transcript? Please clarify.
- 11) In the text (223-225/Figure S7) is it ADAR1/2 mRNA or protein expression levels? Please provide statistics (e.g # of biological replicates, # of technical replicates if applicable, statistical test conducted e.g Wilcoxon test, p=values).

Materials and method

- 1) Please provide information about how microglia, astrocytes, neurons and oligodendrocytes were prepared in regard to the data present in Figure S7 (lines 223-225).
- 2) Was there any variation between the samples regarding the editing sites? Please mention the statistics used throughout the study. Are the identified editing sites were pooled or only the overlapping sites were further investigated?

Discussion

We have the following questions that needed to be addressed and discussed before further consideration.

- 1) It is too often the case that conclusions in the literature do not hold up because the statistics were not robust in the first place. The author wrote: "Unexpectedly, there was not a high correlation between expression of ADAR2 and editing level of coding sites in the pig brain, inconsistent with the situation in human" (lines 349-351). The author explained that could be due to the species-specific regulation or lack of enough statistical power. This conclusion "species-specific regulation" needs to be toned down substantially and instead the statistical issue highlighted. More broadly, the statistical approach, considerations, and limitations of the study and the implications of that, need a full detailed consideration in the discussion and should be briefly mentioned in the abstract so that any conclusions drawn are taken

appropriately in context.

2) Later the authors suggested that pig could be potentially an attractive model of ALS considering the function of ADAR2 mediated editing of GRIA2 RNA. This seems to be overstated and without evidence how and why is this likely to be possible other than as a fairly general statement?

3) How, more broadly, will a pig model will overcome the limitations of using a mice model for studying RNA-editing related dysfunction? Please discuss this critically (e.g refer to Overgaard et al., 2018). Also please consider the relationship of data found in the pig to the human and in that consideration what was the age and state of the human samples analysed versus pig?

4) Following on from the above point, please discuss in detail the inconsistent relation between the expression of ADAR2 and editing level of coding sites between the pig (as determined in this study) and human. How this will be considered for using pig as a model of neurodegeneration?

5) As it's a brain-wide RNA-editing landscape author should also discuss about other RNA-editing target in addition to GRIA2 and their implications.

6) Also please consider the implications of region specificity of RNA editing that are well established in the literature, how that may differ in pig and humans and how regional differences might impact the overall averaged results obtained in whole brain, as assessed in this study

7) It would be important to discuss about new RNA-editing sites observed in this study.

8) The overall study delineates the evolutionary importance of RNA-editing. Discussion on RNA-editing profile comparison between mouse, pig and human would greatly increase the importance of the findings.

9) It is very important to also look into gene expression differences to understand if there are any correlations between RNA-editing sites and gene expression.

10) Are there any correlations between the RNA editing sites and gene expression?

11) 82.79% of all A-to-I editing sites were located in SINEs, this should be further discussed in terms of functional importance (please refer to lines 102-103).

Reviewers' comments:

Reviewer #1 (Remarks to the Author):

This study revealed the RNA editing events in the pig brain for the first time, and made a good supplement to the research on RNA editing in the mammalian brain. However, existing studies have revealed the situation of RNA editing events in the human brain, which greatly reduces the novelty of the research. The RNA editing sites in some key genes require to confirm their authenticity and functions.

1. There are many softwares that have been published to identify RNA editing events. In addition to RES-Scanner, authors can consider choosing other softwares or processes for RNA editing calling, and then compare the consistency of the results from different processes.

RE: Another tool *REDtools* was used for RNA editing calling. Most of RNA editing sites identified by two tools were overlapped. The editing level of common sites discovered by two tools showed a high degree of correlation (shown as follows).

2. What are the effects of the genes with the highest editing level or editing frequency in different regions on regional function, and how is the expression level of these genes?

RE: Highly edited events, especially those located in CDS or 3-UTRs, are potentially functional. Thus, we explored CDS and 3-UTRs residing sites that are highly edited (>75%) in each brain region. The genes with highly edited sites located in CDS or 3-UTR were also investigated. Gene ontology analysis revealed a number of biological processes and molecular functions specific or shared across brain regions. For example, genes with highly edited sites in olfactory bulb, cerebral cortex, amygdala, thalamus, midbrain, pons and medulla, and cerebellum, were enriched in AMPA glutamate receptor activity (shown as follows).

Furthermore, we examined expression levels of genes harboring highly (>75%) or lowly edited (<25%) sites. We found that the expression of genes with highly edited CDS-residing sites tended to be lower than that of genes with lowly edited CDS-residing sites (shown as follows). However, there were not differences in expression between genes with highly and lowly 3-UTRs residing sites.

3. What are the functions of genes with conservative editing sites or with non-conservative sites among species?

RE: Here, we investigated genes with conserved and non-conserved recoding sites. Some genes,

such as GRIK1, GRIK2 and GABRA3, harbored both conserved and non-conserved recoding sites. Ten genes, including receptor genes (GRIA2, GRIA3 and GRIA4), ion channel-related genes (KCNMA1), ion transport-related genes (UNC80 and TMEM63B) and other genes (ADCY6, CPSF6, RICTOR and XKR6), harbored conserved recoding sites solely. The genes with non-conserved recoding sites were mainly in categories related to nucleus, extracellular exosome and RNA binding.

Is there a difference in editing levels between conservative editing sites and non-conservative editing sites?

Re: To investigate whether there are any differences in editing level between conserved and non-conserved sites, we focused on recoding sites. Our analysis revealed that conserved recoding sites were more edited than non-conserved recoding sites in pig brain (shown as follows), in line with earlier observation in mouse.

Does the editing level of conservative editing sites differ among species?

Re: The cross-species comparison of each conserved site was performed across six brain regions, including cerebral cortex, amygdala, hippocampal formation, hypothalamus, cerebellum and spinal cord. Most of the conserved sites were unbiased edited in the two species (shown as follows).

4. How about the RNA editing status of species-specific genes or differentially expressed genes between human and pig?

Re: The editing status of recoding sites in genes showing pig-biased or unbiased expression was explored. A total of 122 human RNA-seq read count data obtained from GTEx Portal were used for cross-species gene expression analysis. Only the one-to-one orthologous genes (n=16538) were taken into account. Differential gene expression analysis between pig and human was performed across six brain regions, including cerebral cortex, amygdala, hippocampal formation, hypothalamus, cerebellum and spinal cord. The pig-biased genes were defined as genes showing 4-fold higher in ≥ 5 regions in pig than in human. No significant difference in recoding editing was observed between pig-biased and unbiased genes (shown as follows).

5. In addition to conserved RNA editing sites, species-specific RNA editing events with high editing level also need attention, because specific high-level editing sites may cause differences in brain function and structure between species. So, please provide more comparisons.

Re: A number of non-conserved recoding sites which were highly edited ($\geq 75\%$) in different brain regions were analyzed and discussed (some were shown as follows). Out of the genes harboring highly edited non-conserved sites, some were potentially important for brain function. For example, UNC13A and UNC13C were involved in chemical synaptic transmission.

6. It is recommended to verify the authenticity and functions of the RNA editing sites of some important genes.

Re: We have now included RNA-seq data from induced porcine neuron from fibroblasts. The recoding sites located in some important genes, such as GRIA2 (Q607R), GRIA2 (R764G), GRIK2 (I518V), GRIK2 (Y522C), and GRIK2 (Q572R), were confirmed by the porcine neuron RNA-seq data, shown in Fig. S14.

7. More experiments may be needed to prove that RNA editing plays an important role in pig brains, and more explanations on the role of RNA editing on brain function are also needed.

Re: As the fundamental units of the pig brain, neurons, of which RNA editing remains little known. Are there any differences in editing between porcine neurons and non-neuronal cells? To address this issue, the RNA-seq data from a previous study was involved in the analysis. In that study, the porcine fibroblasts were directly converted into induced neurons, which expressed common neuronal markers. Our analysis revealed an increase in overall editing from fibroblasts to induced neurons, supporting the idea that RNA editing is essential for neuronal function.

Our analysis of porcine neuron RNA-seq data also revealed an increase in editing level of some recoding sites, such as IGFBP7 (K95R), IGFBP7 (R78G), CYFIP2 (K>E) and NOVA1 (S>G), during conversion of porcine fibroblasts into induced neurons.

Reviewer #2 (Remarks to the Author):

This is a highly interesting and very well written resource manuscript covering A-to-I RNA editing sites in 30 anatomically defined subregions of the porcine brain. The RNA editing sites were identified de novo by a combination of RNA-seq and whole genome sequencing. Given the role of altered RNA editing in neurological disorders and the growing importance of the pig as a model organism, this manuscript will receive a lot of attention and help to design translational pig models for neurological diseases. The manuscript is an important complementation to corresponding studies in human and mouse.

Minor: line 351 “Whether” instead of “Weather”

Re: Typo has been revised.

Reviewer #3 (Remarks to the Author):

In their current study entitled "A porcine brain-wide RNA editing landscape" Huang et al., described a porcine brain-wide RNA landscape which provides a rich resource to better understand the evolutionary importance of post-transcriptional RNA editing. They identified a total of 682,037 A-to-I RNA editing sites in 30 different anatomically distinct brain subregions in pig brain. Importantly, 97% of the RNA editing sites were not previously identified and highly conserved recoding events are found in neurotransmitter receptors demonstrating an evolutionary importance of RNA-editing process on brain function. Furthermore, these studies reconfirm A-to-I editing of GluA2 RNA is critical for brain function across different mammalian species. Over all these studies provide a framework to investigate the role of previously unidentified RNA-editing in human brain function. The manuscript is generally well written, with the hypothesis and experimental paradigms well justified and the results of the study not overstated. If the authors are able to provide more clarification, primarily in the discussion section and minor changes in the description of figure legends, statistics used in the methodology section then manuscript will be suitable for publication.

Specific comments, with recommendations for addressing each comment

1) In the text (lines 94-97), the author showed 70 % of editing sites are located within the protein coding sequences. Notably, 67.4% editing sites in the introns. While the finding is very interesting, looking at the Figure 1a, it is hard to understand the % of editing sites in the graph. Is the quantification was determined in relation to total 100% editing sites? Please clarify.

Re: The figure has been revised. It was relative to total number of RNA editing sites identified in this study (n=682,037).

Fig. 1

2) Identification of clustering tendency of the editing sites is very informative. The author mentioned (lines 116-117) that clustering studies were previously, carried out in ant. It is critical to mention whether this kind of studies were performed in mice or rat.

Re: The mice have been mentioned.

3) Is there in any studies in the literature that suggest clustering if 3 sites are located in a 100bp sliding window (lines 118-119)?

Re: In a previous study, Li, Q. *et al.* defined a 100 bp sliding window with ≥ 3 editing sites as a cluster. (See supplementary Methods, Li, Q. *et al.* Caste-specific RNA editomes in the leaf-cutting ant *Acromyrmex echinator*. Nature communications 5, 4943, doi:10.1038/ncomms5943 (2014))

4) Please provide the information about statistics in the legend for Figure 2b and 2c. (e.g # biological replicates, # of technical replicates if applicable, statistical test conducted e.g Wilcoxon test, p=values)

Re: The legend has been revised. The number of biological replicates in each main region is as follows: OLF (n=5), CTX (n=31), AMY (n=4), HPF (n=16), BG (n=16), HY (n=4), TH (n=4), MB (n=15), PM (n=8), CB (n=4), CC (n=4) and SC (n=8). The dashed line denotes the median value of all regions. The significance of differences between overall editing level of each region and that of all regions was assessed by Wilcoxon test (* $P \leq 0.05$, and ** $P \leq 0.01$).

5) Which expression levels were measured (lines 209-211)? Is it mRNA or protein expression levels? Expression levels were normalized against which reference gene or protein? Kindly, provide the statistical information for graphs of ADAR1, ADAR2 and ADAR3 expression in Figure 3a.

Re: It is the level of mRNA expression. The expression levels were normalized against normalization factor using DEseq. Firstly, the geometric mean count of each gene across all samples was calculated. Secondly, the normalization factor (also known as size factor) for a given sample was determined by median ratio of gene counts relative to geometric mean per gene. Lastly, the raw count of each gene was divided by normalization factor in each sample, to generate normalized count value (mRNA normalized expression).

The statistical information is described in the legend and the figure has been revised. Bar denotes Median + IQR for biological replicates in each main region. The dashed line denotes the median value of all regions. The significance of differences between mRNA expression of each region and that of all regions was assessed by Wilcoxon test (* $P \leq 0.05$, ** $P \leq 0.01$, *** $P \leq 0.001$, and **** $P \leq 0.0001$).

Fig. 3
a

6) What are these abbreviations PC1 and PC2 in Figure 5a? Please provide the information in figure legend.

Re: The legend has been revised.

7) This figure legends in Figure S1 should describe BWA, HISAT2 instead of using abbreviation only. It will help reader to follow and check the methodology in detail. Also, it would be good to mention the 30 anatomically different regions in the flowchart.

Re: The program BWA and HISAT2 have been described in the legend. The 30 subregions are also described in the flowchart.

8) In Figure S2, please write this is WGS analysis. It would be important to mention which brain regions were analysed in this graph.

Re: The figures and legend have been revised. All 30 subregions are listed, rather than average in pig brain.

9) Kindly clarify (line 138, Figure S3b), whether the number of events is an absolute or relative number.

Re: It is an absolute number. The figure has been revised.

10) In the text (lines 220-221/Figure S6) is it mRNA or protein expression levels (Figure S6)? Is this overall A-to-I editing levels? Or of a particular edited transcript? Please clarify.

Re: Here, we aim to investigate the correlation between the overall A-to-I editing levels and mRNA expression of ADAR1, ADAR2, etc. The text and figures have been revised.

For example,

11) In the text (223-225/Figure S7) is it ADAR1/2 mRNA or protein expression levels? Please provide statistics (e.g # of biological replicates, # of technical replicates if applicable, statistical test conducted e.g Wilcoxon test, p=values).

Re: It is mRNA expression score. The statistics information has been described in the figures.

Materials and method

1) Please provide information about how microglia, astrocytes, neurons and oligodendrocytes were prepared in regard to the data present in Figure S7 (lines 223-225).

Re: The mRNA expression resource of human neurons and non-neuronal cells has been described in Materials and Methods. The mRNA expression scores of neurons, microglia, astrocytes, and oligodendrocytes from 20 human brain regions, which are based on co-expression analysis, were obtained from <http://oldhamlab.ctec.ucsf.edu/>. The significance of differences between expression scores of ADAR1 or ADAR2 in neurons and non-neuronal cells was assessed by Wilcoxon test.

2) Was there any variation between the samples regarding the editing sites? Please mention the statistics used throughout the study. Are the identified editing sites were pooled or only the overlapping sites were further investigated?

Re: Previous study reported that the number of edited sites identified could increase as more sequencing data are generated. Considering variations in sequencing depth among samples, the editing sites identified in samples from the same subregion/region were pooled for further investigations. The statistics used throughout this study has been described in Materials and

Methods.

Discussion

We have the following questions that needed to be addressed and discussed before further consideration.

1) It is too often the case that conclusions in the literature do not hold up because the statistics were not robust in the first place. The author wrote: “Unexpectedly, there was not a high correlation between expression of ADAR2 and editing level of coding sites in the pig brain, inconsistent with the situation in human” (lines 349-351). The author explained that could be due to the species-specific regulation or lack of enough statistical power. This conclusion “species-specific regulation” needs to be toned down substantially and instead the statistical issue highlighted. More broadly, the statistical approach, considerations, and limitations of the study and the implications of that, need a full detailed consideration in the discussion and should be briefly mentioned in the abstract so that any conclusions drawn are taken appropriately in context.

Re: The limitations in this study have been mentioned in the abstract briefly and described in detailed in the discussion section.

2) Later the authors suggested that pig could be potentially an attractive model of ALS considering the function of ADAR2 mediated editing of GRIA2 RNA. This seems to be overstated and without evidence how and why is this likely to be possible other than as a fairly general statement?

Re: The text has been deleted.

3) How, more broadly, will a pig model will overcome the limitations of using a mice model for studying RNA-editing related dysfunction? Please discuss this critically (e.g refer to Overgaard et al., 2018). Also please consider the relationship of data found in the pig to the human and in that consideration what was the age and state of the human samples analysed versus pig?

Re: To investigate the underlying mechanisms of human disease, mouse is one of the primary model organisms. However, anatomical differences between mouse and human brain are needed to be considered. For example, mouse lacks gyri and sulci in the cerebrum, unlike human or pig. In addition, many genes related to neurotransmission are subject to RNA editing. A higher overall correlation between pig and human brain for expression of genes related to neurotransmission was observed, as compared with that between mouse and human brain.

We failed to reveal high correlation between ADAR2 expression and editing of coding sites in pig brain. However, most of conserved CDS-residing sites, including many physiologically important editing events, were unbiased edited between pig and human brain. Based on observations above, pig may be an alternative choice for studying dysregulated RNA editing associated with human neurological disorders.

Potential data biases caused by difference in developmental stage between species have been discussed in the limitation section.

4) Following on from the above point, please discuss in detail the inconsistent relation between the expression of ADAR2 and editing level of coding sites between the pig (as determined in this study) and human. How this will be considered for using pig as a model of neurodegeneration?

Re: We failed to reveal high correlation between ADAR2 expression and editing of coding sites in pig brain. However, most of conserved CDS-residing sites, including many physiologically important editing events, were unbiased edited between pig and human brain. Pig may be an alternative choice for studying dysregulated RNA editing associated with human neurological disorders.

5) As it's a brain-wide RNA-editing landscape author should also discuss about other RNA-editing target in addition to GRIA2 and their implications.

Re: More editing events have been included for discussions. Many physiologically important editing events in mammals were unbiased edited between pig and human brain. For example, out of unbiased edited sites found in cerebral cortex, GRIA2 (Q607R) and GRIK2 (Q572R) are involved in change in Ca²⁺ permeability. GRIA2 (R764G), GRIA3 (R775G) and GRIA4 (R765G) are involved in change in receptor desensitization. GABRA3 (I342M) is involved in receptor trafficking.

6) Also please consider the implications of region specificity of RNA editing that are well established in the literature, how that may differ in pig and humans and how regional differences might impact the overall averaged results obtained in whole brain, as assessed in this study

Re: The regional variations in RNA editing have been observed in the human (previous study) and pig brain (this study), indicating that RNA editing is spatially regulated in mammalian brain. The similarities in regional RNA editing profiles were seen in the two species. The overall editing levels of CDS-residing sites tended to be similar across brain regions, whereas editing levels of those sites in repetitive region were more likely to be different. For example, the repetitive sites in cerebellum were edited more in both human and pig brain. Consistently, the region-specific analysis on editing sites also revealed that pig cerebellum harbored the largest number of region-specific edited sites.

7) It would be important to discuss about new RNA-editing sites observed in this study.

Re: Many functionally important editing events were not described in previous database. The importance of novel sites identified in this study have been discussed in the "Novel editing sites" section.

8) The overall study delineates the evolutionary importance of RNA-editing. Discussion on RNA-editing profile comparison between mouse, pig and human would greatly increase the importance of the findings.

Re: The similarities among human, mouse and pig RNA editing profiles have been discussed. In all

three species, most A-to-I editing occur within repetitive elements, especially SINEs, e.g. Alu in human, B1 in mouse and PRE in pig. Another similarity among human, mouse and pig is the fact that many conserved recoding events occur in neuronal genes.

9) It is very important to also look into gene expression differences to understand if there are any correlations between RNA-editing sites and gene expression.

10) Are there any correlations between the RNA editing sites and gene expression?

Re: Point 9 and 10:

The relationship between RNA editing and mRNA expression was investigated. Our analysis revealed that there was weak linear relationship for most of RNA editing sites and genes analyzed.

11) 82.79% of all A-to-I editing sites were located in SINEs, this should be further discussed in terms of functional importance (please refer to lines 102-103).

Re: The functional importance of editing has been described in the discussion section in more detail.

Reviewers' Comments:

Reviewer #1:

Remarks to the Author:

All issues have been solved. Considering its scientific value and solid experiments, I believe its current version will be suitable to be published in the journal.

Reviewer #2:

Remarks to the Author:

Thank you for the extensive revision of the manuscript.

Reviewer #1 (Remarks to the Author):

All issues have been solved. Considering its scientific value and solid experiments, I believe its current version will be suitable to be published in the journal.

Reviewer #2 (Remarks to the Author):

Thank you for the extensive revision of the manuscript.

Re: We thank the reviewers for taking their time to thoroughly evaluate our study and the manuscript. All the valuable suggestions in both experiments and analyses suggested by the reviewers have substantially improved our study and the final version of the manuscript. We also thank their recommendation of publication.